# Activity of a Recombinant Chitinase of the *Atta sexdens* Ant on Different Forms of Chitin and Its Fungicidal Effect against *Lasiodiplodia theobromae*

**DOI:** 10.3390/polym16040529

**Published:** 2024-02-15

**Authors:** Katia Celina Santos Correa, William Marcondes Facchinatto, Filipe Biagioni Habitzreuter, Gabriel Henrique Ribeiro, Lucas Gomes Rodrigues, Kelli Cristina Micocci, Sérgio Paulo Campana-Filho, Luiz Alberto Colnago, Dulce Helena Ferreira Souza

**Affiliations:** 1Department of Chemistry, Federal University of Sao Carlos, 13565-905 Sao Carlos, Brazil; katiacorrea12@gmail.com (K.C.S.C.); lucasgomes@estudante.ufscar.br (L.G.R.); kelli.micocci@gmail.com (K.C.M.); 2Aveiro Institute of Materials, CICECO, Department of Chemistry, University of Aveiro, St. Santiago, 3810-193 Aveiro, Portugal; williamfacchinatto@alumni.usp.br; 3Sao Carlos Institute of Chemistry, University of Sao Paulo, Ave. Trabalhador Sao-carlense 400, 13560-590 Sao Carlos, Brazil; filipeh@usp.br (F.B.H.); scampana@iqsc.usp.br (S.P.C.-F.); 4Brazilian Corporation for Agricultural Research, Embrapa Instrumentation, St. XV de Novembro 1452, 13560-970 Sao Carlos, Brazil; gabrielhenri10@hotmail.com (G.H.R.); luiz.colnago@embrapa.br (L.A.C.)

**Keywords:** insect chitinase, chitin, fungicide

## Abstract

This study evaluates the activity of a recombinant chitinase from the leaf-cutting ant *Atta sexdens* (AsChtII-C4B1) against colloidal and solid α- and β-chitin substrates. ^1^H NMR analyses of the reaction media showed the formation of N-acetylglucosamine (GlcNAc) as the hydrolysis product. Viscometry analyses revealed a reduction in the viscosity of chitin solutions, indicating that the enzyme decreases their molecular masses. Both solid state ^13^C NMR and XRD analyses showed minor differences in chitin crystallinity pre- and post-reaction, indicative of partial hydrolysis under the studied conditions, resulting in the formation of GlcNAc and a reduction in molecular mass. However, the enzyme was unable to completely degrade the chitin samples, as they retained most of their solid-state structure. It was also observed that the enzyme acts progressively and with a greater activity on α-chitin than on β-chitin. AsChtII-C4B1 significantly changed the hyphae of the phytopathogenic fungus *Lasiodiplodia theobromae*, hindering its growth in both solid and liquid media and reducing its dry biomass by approximately 61%. The results demonstrate that AsChtII-C4B1 could be applied as an agent for the bioproduction of chitin derivatives and as a potential antifungal agent.

## 1. Introduction

Chitin is a polysaccharide that occurs abundantly in nature and is a structural component of many organisms, such as mollusks, fungi, and arthropods [1]. Composed of a linear chain of N-acetyl-D-glucosamine (GlcNAc) monomers, linked by β-(1-4) glycosidic bonds [2], chitin occurs in nature as three polymorphs called α-, β-, and γ-chitin [3]. These polymorphs have different arrangements of their polymer chains in the crystalline domains, which results in marked differences in their physicochemical properties, such as their crystallinity and swelling capacity. The polymorphs also differ in their degree of hydration, the size of the unit cell, and the number of chitin chains per unit cell [4]. The molecular organization of chitin involves macromolecules that interact with other elements either covalently or supramolecularly, which defines many of the functions in the organism, ranging from growth and mechanical resistance to defense against microorganisms and diseases [5]. The microfibrils combine with sugars, proteins, glycoproteins, and proteoglycans to form cell walls in fungi, as well as the arthropod cuticles and peritrophic matrices present in crustaceans and insects, respectively [4,6].

Due to its abundance and also to the numerous possibilities for carrying out chemical modifications, which can result in chitin whiskers via acidolysis, chitosan via N-deacetylation, oligomers and GlcNAc units via hydrolysis, and chitosan, chitin derivatives have enormous potential for use in the pharmaceutical, cosmetics, and nutritional supplement industries [7,8] and have also aroused the interest of the scientific community [9]. The characteristics of this material include its biodegradability, biocompatibility, and antioxidant [10] and antibacterial activities [11], and thus it has been used in the food and health industries [12]. 

Chemical, enzymatic, and physicochemical treatments can be used to make chemical changes to chitin’s structure, including reducing the average molecular mass via depolymerization [13]. Depolymerization via chemical treatment is the most traditional method and is mainly performed via deacetylation/degradation through alkaline/acid treatment and the introduction of new chemical groups using specific solvents. The modification effect depends on the established conditions, including the type of reagent, time, temperature, and pH [14,15]. Chemical methods are efficient, but it is necessary to rethink their use because of the formation of polluting residues [16]. Physical methods are employed to reduce the use of chemical reagents, such as the use of ultrasound, which has been applied to provoke modifications in the structure of chitin, providing porosity and reducing the size of fibers or particles. However, the use of these methods on an industrial scale presents complications such as a high energy consumption and poor control of the products formed [17].

Thus, enzymatic hydrolysis, which has a low energy consumption, reduces the generation of polluting residues, and enables greater control of the products formed, is a very promising methodology and is being increasingly employed [18]. Enzymatic hydrolysis processes are generally conducted between 30 and 60 °C, with a pH ranging from 4 to 12 and a duration of a few hours [17,18,19,20]. The enzymes that catalyze hydrolysis reactions of chitin, called chitinolytic enzymes, belong to the family of glycosyl hydrolases (GHs) [21]. According to the carbohydrate-active enzyme database (CAZy), chitinolytic enzymes are classified into the families GH18, GH19, GH23, and GH48, with the difference between the GHs being the composition of their amino acids and catalytic properties [22]. Chitinases of the GH18 family are present in almost all organisms, including plants and mammals, and are classified according to the type of cleavage they promote. Endo-chitinases catalyze the internal hydrolysis of chitin chains at random positions, whereas exo-chitinases hydrolyze the chitin chain at its terminal, whether at the reducing or non-reducing end [13].

In organisms, chitinases are involved in tissue degradation, developmental regulation, pathogenicity, and immunological defense. Although the most extensive function of GH18 chitinases is to degrade endogenous chitin, many microorganisms produce them to utilize chitin as a nutritional source [23]. Insect chitinolytic enzymes, for example, have been identified as potential biopesticides against organisms that contain chitin in vital structures, such as the peritrophic membrane or cuticle of insects; eggshells; nematode sheaths; and the cell walls of pathogenic fungi [24]. For instance, a chitinase from *Bombyx mori* was evaluated for its potential use as a biopesticide against the *Monochamus alternatus* beetle. The oral ingestion of chitinase induced modifications in the beetle’s peritrophic membrane chitin, leading to a reduced body weight and mortality [25]. A chitinase from *Ostrinia furnacalis* showed activity against phytopathogens such as *Fusarium graminearum*, *Botrytis cinerea*, *Rhizoctonia solani*, *Phytophthora capsici*, and *Colletotrichum gloeosporiodes* [26].

Recently, our research group expressed in *Pichia pastoris* a chitinase from the leaf-cutting ant *Atta sexdens* (AsChtII-C4B1), which consists of a catalytic domain and a chitin-binding domain (CBM) belonging to the GH18 family of GHs [27]. AsChtII-C4B1 exhibited larvicidal activity against the tested model *Spodoptera frugiperda* and fungicidal activity against human pathogenic fungi *Candida albicans* and *Aspergillus fumigatus*. In this present study, we investigate the hydrolytic action of AsChtII-C4B1 on different forms of chitin (α- and β-chitin) and evaluate the activity of the recombinant chitinase against the phytopathogenic fungus *Lasiodiplodia theobromae* a cause of severe losses in agricultural production.

## 2. Materials and Methods

### 2.1. Materials

The fungus *Lasiodiplodia theobromae* (CDA 1169), isolated from the stem of soursop (*Annona muricata*), Jaiba, MG, Brazil, was acquired from the Mycological Collection of the Federal University of Pernambuco, Brazil, and maintained in sterile water according to the CASTELLANI method [28] at room temperature until use. Subsequently, the fungus was inoculated at 28 °C in a plate containing potato dextrose agar (PDA).

The recombinant chitinase AsChtII-C4B1 was obtained as reported in the literature, being expressed into the extracellular medium in a *Picchia pastoris* system and purified on a nickel resin affinity column [27].

### 2.2. Methods

#### 2.2.1. Evaluation of Enzyme Activity against Various Forms of Chitin

##### α- and β-Chitin Substrates

β-chitin was extracted from the gladii of squids (*Doryteuthis* spp.) following the methodology described in the literature [29], ground in a knife mill equipped with a 1 mm sieve, and then separated into different powder fractions with average diameters (d) varying from <0.125 to >0.425 mm [30]. α-chitin, isolated from crab shells, was purchased from SIGMA ALDRICH (St. Louis, MI, USA). α-chitinase and colloidal β-chitin were prepared according to a method described in the literature [31] and used at a concentration of 5% (*w/v*) in citrate-phosphate buffer, pH 5.0 [32].

##### Enzymatic Activity against Various TYPES of Chitin

For the reaction medium, 1.0 mL of purified protein (0.713 mg) was added to individual tubes containing the solid substrates βQ125SE (β-chitin d < 125 mm) and βQ425SE (β-chitin d > 0.425 mm), both at a concentration of 5% (*w/v*) in citrate-phosphate buffer, pH 5.0, totaling a final volume of 2 mL. The colloidal β-chitin (βQ125CE and βQ425CE) and α-chitin (αCenz) substrates underwent the same enzymatic treatment. The reaction medium contained in the tubes was stirred at 250 rpm at 55 °C for 48 h. The solution was centrifuged at 10,000× *g* at 4 °C for 40 min and the pellets were homogenized in 5 mL of MilliQ H_2_O (St. Louis, MI, USA). After this step, dialysis of the reaction medium was performed using the dialysis membrane (3.5K MWCO) against 600 mL of ultrapure H_2_O at 4 °C overnight.

The same procedure previously described, but in the absence of the enzyme, was conducted using colloidal α-chitin at 55 °C (here called CT55) and at 25 °C (here called chitin CT25).

#### 2.2.2. Evaluation of GlcNAc Production via ^1^H Nuclear Magnetic Resonance (NMR) Spectroscopy Post Enzymatic Reaction

Aliquots of 350 µL from the samples (supernatant) of the enzymatic reaction against colloidal α- and β-chitin substrates were diluted in 250 µL of an internal standard solution of sodium 3-trimethylsilylpropionate-d4 (0.50 mM, TMSP in D_2_O), used as an internal standard at 0.00 ppm. The samples were transferred to a standard 5 mm NMR tube for analysis.

The 1D and 2D NMR experiments of the samples were performed at 25 °C using a Bruker 14.1 Tesla instrument, AVANCE III (Billerica, MA, USA), equipped with a 5 mm PABBO (Broad Band Observe) direct detection probe with ATMA^®^ (Automatic Tuning Matching Adjustment) and a BCU-I variable temperature unit. The ^1^H NMR spectra were acquired using a pre-saturation solvent suppression pulse sequence of the water signal (here called noesypr1d), Bruker TopSpin, with a field gradient and with water signal suppression by irradiation at the frequency of 2822.04 Hz (O1). The conditions were as follows: 64 averages (ns), 4 dummy scans (ds), 65,536 data points during acquisition (td), a spectral window (sw) of 20.03 ppm, a receiver gain (rg) of 80.6, a 90° pulse of 11.850 µs, an acquisition time between each acquisition (aq) of 2.73 s, and a 5 ms mixing time (d8). The ^1^H NMR spectra were referenced through the TMSP-d4 signal at 0.0 ppm. To support the assignment of the compound of interest, 2D NMR experiments were conducted, such as ^1^H-^1^H COSY.

The ^1^H NMR spectra were processed using TopSpinTM 3.6.1 software (Bruker, Biospin, Ettlingen, Germany). To determine the product of interest, N-acetylglucosamine (GlcNAc), a database query was carried out (human metabolome; HMDB—Human Metabolome Database; N-acetylglucosamine (HMDB0000215)). The GlcNAc assignments were confirmed by an analysis of 2D NMR correlation maps. The compound was quantified in the ^1^H NMR spectra using Chenomx NMR Suite 8.4 software (Chenomix Inc, Edmonton, Edmonton, AB, Canada). ^1^H-^13^C HMBC, Jres, was performed on a selected sample.

#### 2.2.3. Analysis of the Morphology of Various Types of Chitin

##### Solid-State ^13^C Nuclear Magnetic Resonance (NMR) Spectroscopy

The solid-state NMR ^13^C spectra of the chitin samples (βQ125SE, βQ425SE, βQ125CE, βQ425CE, CEnz, αQCT25, and αQCT55 °C) were obtained at 25 ± 1 °C using an Advance 400 spectrometer (Bruker) coupled to a 4 mm dual-resonance probe with magic-angle spinning (MAS), operating at 100.5 MHz for the carbon nucleus and 400 MHz for the hydrogen nucleus. The average degree of acetylation (GA) was calculated and short-range molecular ordering was assessed. The short-range crystallinity index (CrI_SR_) was calculated from the fitting of signals from carbon C4 and C6 following the deconvolution method proposed in the literature [33].

##### X-ray Diffraction (XRD)

XRD patterns of the chitin samples (βQ125SE, βQ425SE, βQ125CE, βQ425CE, Cenz, CT25, and C55 °C) were acquired using an AXS D8 Advance diffractometer (Bruker; Billerica, Billerica, MA, USA) with a Cu Kχ radiation source (λ = 0.1548 nm). Measurements were performed in the range 5° < 2θ < 50° at a scan speed of 5° min. The crystallinity indices (ICr) were estimated by subtracting the contribution of the amorphous region (A_am_) by fitting a cubic spline curve from the total diffraction pattern area (A_tot_). Data treatment was conducted using Microcal Origin 2020 software [34]. In addition, the apparent crystalline dimensions (L_hkl_) referring to (020)_h_ and (200)_h_ reflections were calculated by applying the Scherrer relation through the width at half-heights (FWHM) of diffraction peaks at 2θ ≈ 8.3° and ≈ 19.7°, respectively. Data treatment was conducted by fitting Lorentzian curves, as described in previous studies [30,33].

##### Capillary Viscometry in Dilute Regime

The viscosity average molecular masses $\left(M_{v}\right)$ of the chitin samples were determined from their intrinsic viscosities (η) and average degrees of acetylation (GA). To determine η, pre- and post-enzymatic treatment chitins were dissolved in N,N-dimethylacetamide (DMAc) containing 5% LiCl (*w/v*). The samples were dissolved in DMAc/LiCl at room temperature for 24 h and then filtered under positive pressure (0.45 um). The flow times in a glass capillary (ϕ = 0.84 mm) were determine at 25.00 ± 0.01 ^0^C using an AVS-360 viscometer connected to an AVS-20 automatic burette, both from Schott-Geräte (Mainz, Germany). From the extrapolation of the line obtained at infinite dilution, it is possible to determine the η value of the polymer, which is necessary to calculate $M_{v}$, as previously established in the literature [35,36]. The values of $M_{v}$, GA, and the molecular masses (g/mol) of GlcN and GlcNAc units were used to calculate the average degree of polymerization ${(GP}_{v})$, as indicated in previous studies [31,34]. The assays were conducted in triplicate.

### 2.3. Evaluation of the AsChtII-C4B1 Enzyme Activity on the Growth of the Fungus L. theobromae

#### 2.3.1. Assays of Activity and Thermostability of AsChtII-C4B1

The chitinase activity in biological assays was determined by the 3,5-dinitrosalicylic acid (DNS) method (Sigma-Aldrich; St. Louis, MI, USA), according to the literature [37,38], using colloidal α-chitin as a substrate. Briefly, 200 µL (0.138 mg) of the purified enzyme was homogenized with 200 µL of 5% (*w/v*) colloidal α-chitin in citrate-phosphate buffer, pH 5.0. The solution was incubated at 55 °C with shaking at 250 rpm for 1 h. After that, 400 µL of DNS was added to the reaction. The reaction mixture was heated at 100 °C for 10 min and then cooled to −20 °C for 5 min. The solution was centrifuged at 10,000× *g* for 5 min and the supernatant was subjected to absorbance measurements at 540 nm in a spectrophotometer (BIOMATE 160; Mettler Toledo; Langacher; Greifensee, Switzerland). The blank control mixture (without the enzyme) underwent the same treatment and was used to zero the equipment. The results obtained were compared with a standard curve of (GlcNAc) ranging from 0.1 to 1 mg mL^−1^.

The enzymatic activity was also assessed on both colloidal and solid α- and β-chitin substrates by varying the incubation time (1–72 h), measuring the enzyme concentration in the reaction solution via the Bradford method [39], using BSA as a standard. The percentage of enzyme in the solution was calculated from the difference between the total enzyme concentration and the remaining enzyme concentration in the supernatant during the assay period. All experiments were conducted in triplicate.

#### 2.3.2. Antifungal Assays

From the fungus cultivated on a Petri dish (90 cm × 15 mm) in PDA medium (Potato Dextrose Agar M096, HIMEDIA), an 8 mm halo was removed and placed in 10 mL of liquid medium (5.6 g L^−1^ (NH4)_2_SO_4_, 4 g L^−1^ KH_2_PO_4_, 0.6 g L^−1^ MgSO_4_.7H_2_O, 1.8 g L^−1^ peptone, 0.5 g L^−1^ yeast extract, 0.02 g L^−1^ MnSO_4_.H_2_O, 0.002 g L^−1^ ZnSO_4_.7H_2_O, 0.04 g L^−1^ CoCl_2_.6H_2_O) [40]. The culture medium was incubated at 28 °C with constant agitation at 150 rpm for 72 h. Subsequently, purified chitinase (0.713 mg) was added to the medium and the culture was maintained for another 72 h under the same conditions to verify the fungal mycelial growth.

The fungus’s dry mass was calculated after cultivation under the previously described conditions at three different times: 24, 48, and 72 h. To this end, after the desired time, the culture medium was centrifuged, rinsed with Milli-Q water, dried at 60 °C, and weighed.

In another experiment, an 8 mm mycelial sample was removed from the Petri dish cultivated with the fungus, plated on a new Petri dish, and a solution containing 0.713 mg of AsChtII-C4B1 was dripped onto the halo. The same procedure was conducted in the absence of the enzyme (positive growth control) and the presence of 2 µg mL^−1^ of the commercial fungicide Amphotericin-B (negative growth control). The results are expressed as the mean of three replicates of three independent experiments with the indicated standard deviations.

#### 2.3.3. Analysis of Biological Samples via Scanning Electron Microscopy (SEM)

The mycelial samples from the liquid medium experiment mentioned in Section 2.3.2. were incubated in a Karnovsky solution (4% paraformaldehyde, 5% glutaraldehyde, 0.05% CaCl_2_) [41] at room temperature for 24 h to fix and preserve the biological material. Subsequently, the Karnovsky solution was discarded and the samples were dehydrated by varying the percentage of acetone every 10 min (30, 50, 70, 90, and 100%), with the 100% step performed 3 times every 10 min. The mycelia were lyophilized and coated with a gold layer for SEM analysis to observe possible morphological changes in the fungal hyphae. Micrographs were obtained using JEOL equipment, model JSM 6510 (Tokio; Tokyo, Japan), with an electron acceleration voltage of 5 kV and a working distance (WD) of 10 mm.

## 3. Results and Discussion

### 3.1. Activity of Recombinant Chitinase on Various Substrates

The recombinant chitinase AsChtII-C4B1 contains a catalytic domain and a carbohydrate-binding module (CBM), which has been reported to assist in anchoring the enzyme to the insoluble substrate through the interaction of conserved aromatic residues, breaking the crystalline structure of the substrate, resulting in the formation of free chain ends [23,42]. This enzyme shows activity against colloidal α-chitin and has been shown to have fungicidal and larvicidal activity [27]; it was inferred that it might also act on solid-state chitin. Thus, this study assessed the activity of the enzyme against solid α-chitin and colloidal and solid β-chitin.

Considering the modes of enzymatic activity, GH18 chitinases can be divided into processive and non-processive chitinases. Processive chitinases can slide along the substrate chain and continue hydrolysis without the enzyme, detaching from the chitin chain after each catalytic event, thus producing soluble reducing ends in contrast to non-processive chitinases. In general, exo-chitinases are processive enzymes, whereas endo-chitinases are non-processive [43].

To infer the mode of action of the AsChtII-C4B1 enzyme, the protein concentration in solution was quantified post-reaction (monitored up to 72 h) with α- and β-chitin substrates in both solid and colloidal states (Figure 1). In all experiments, the protein concentration in solution decreased over the reaction duration, suggesting that AsChtII-C4B1 is a processive enzyme, binding to the substrate and catalyzing the cleavage of consecutive bonds without dissociating from it.

Analysis of the free protein concentration demonstrated that solid β-chitin was the most accessible substrate for AsChtII-C4B1 binding, as it showed the lowest concentration in solution at all evaluated reaction times. One hour post-reaction with this chitin, only 34% of the enzyme was free, while in the reaction with solid α-chitin, 74% of the enzyme was free. For colloidal substrates, β-chitin also proved to be more accessible to enzyme binding than α-chitin. The chains of chitin molecules are organized in sheets, strongly held together by hydrogen bonds, and the structure of β-chitin has fewer hydrogen bonds between neighboring chains compared with that of α-chitin, forming less dense fibrils that are more susceptible to swelling and hydrolysis reactions [44]. Thus, the results obtained in this experiment, showing that β-chitin is more susceptible to enzyme binding, can be explained by the structural nature of the substrates.

Studies have shown that the major impediment to enzymatic hydrolysis is the crystallinity of the substrate [7,45]. By synthesizing colloidal chitins, molecules with a lower crystallinity are obtained, increasing the amorphous regions, which are more accessible to enzymatic action. Experiments with α-chitin revealed that the colloidal molecule has a higher concentration of protein bound to the substrate than in the solid state, as expected. However, this behavior was not observed with β-chitin, as the enzyme bound at greater concentrations in the solid form than in the colloidal form. Differences in substrate preferences by chitinases have been associated with the presence/absence of a CBM, with chitinases having a CBM reported as being more efficient at degrading crystalline chitin, while those without a CBM act on less crystalline chitin [22]. AsChtII-C4B1 has a CBM, and this enzyme is capable of catalyzing the cleavage reactions of the substrate bonds in both solid and colloidal forms, other factors, besides CBM, might be involved in the enzyme’s binding to the substrates.

### 3.2. H NMR Spectroscopy of Enzymatic Hydrolysis on α- and β-Chitin Colloidal Substrates

^1^H NMR spectroscopy is a highly specific and sensitive method for quantifying and determining the products resulting from chitin hydrolysis. Since enzymatic hydrolysis is conducted under mild conditions and shows high selectivity in producing, for example, GlcNAc and (GlcNAc)_2_, this technique is useful in assessing chitinase activity, exhibiting an excellent correlation between product concentration and peak integrals [46].

Therefore, the product generated in the reactions of AsChtII-C4B1 with the colloidal α- and β-chitin substrates was identified and quantified via ^1^H NMR spectroscopy (Figure 2). Signals corresponding to the hydrogens of GlcNAc were identified in the spectrum. Their chemical structures and assignments are presented in the Appendix A. To support the assignment of the produced compound, 2D NMR experiments, such as ^1^H-^1^H COSY (Appendix A), were performed. For the quantification of GlcNAc, signals from the hydrogens of the N-acetyl group (5.19 ppm) and the anomeric hydrogens of GlcNAc (2.03 ppm) were selected. The quantities of GlcNAc found were 2.64 and 2.16 mmol L^−1^ for the α- and β-chitin substrates, respectively, showing that the AsChtII-C4B1 enzyme is active on both substrates.

### 3.3. Activity of AsChtII-C4B1 on Different Types of Chitin

The enzyme activity on various types of solid and colloidal chitins was evaluated pre- and post-enzymatic action using the solid-state ^13^C NMR technique. Before treatment, the spectra of βQ425 chitin (β-chitin with particle d > 425 nm) and βQ125 chitin (β-chitin with particle d < 125 nm) showed only one signal around 77 ppm, which is due to carbons C3 and C5. After enzymatic treatment of the solid chitin samples (βQ425SE and βQ125SE) (Figure 3a,b), the signal at 77 ppm showed a shoulder, and the signal at 63 ppm became slightly wider.

In the case of the reaction with the 425 and 125 colloidal β-chitins (βQ425CE and βQ125CE) (Figure 3a,b), the signal around 77 ppm was split into two signals at 73 and 78 ppm, corresponding to carbon C3 and C5 (references). Typically, for β-chitin samples, these signals overlap, while for α-chitin (Figure 3c), they are in slightly different chemical environments and do not overlap [47,48]. The separation of the signals of carbon C5 and C3 indicates a greater structural homogeneity, resulting from the antiparallel arrangement of the α-chitin chains [45]. Thus, it seems plausible to consider that, in the βQ425CE and βQ125CE substrates, new ordered arrangements are formed, resembling those of α-chitin, suggesting an increase in the substrate’s relative crystallinity [7,49]. Conversion from the β-chitin allomorph to α-chitin has been reported in previous studies [50,51]. According to these studies, the process of recrystallization into a more thermodynamically stable morphology can be achieved under conditions of heating/cooling the sample, as well as solubilization in strongly acidic systems. In both cases, the breakdown of microfibrils is ensured, forcing the system to adopt an arrangement that provides greater packing of the chains. In the present case, the separation of carbon C5 and C3 signals appears as the first evidence that recrystallization can also be achieved under enzymatic conditions.

It is worth noting that the profiles of the other spectral signals in Figure 3 do not indicate a significant alteration in the structure. In particular, slight changes quantified from the signals of carbons C4 and C6 indicate a small increase in the short-range crystallinity index (CrI_SR_) in the colloidal substrates compared with the solid substrates treated with the enzyme, but both are lower than the values of the starting substrates. Indeed, the enzyme’s action on any polymorph proves to be efficient in breaking the crystalline domains. It is also worth highlighting that the effect of different processing temperatures on the commercial reference substrate led to an equivalent decrease in CrI_SR_ values compared with those achieved via enzymatic treatment (Figure 3c).

The XRD patterns of the α- and β-chitin samples are illustrated in Figure 4. Even with enzymatic treatment, the solid substrates (βQ425SE and βQ125SE) exhibit profiles similar to those of the original substrates (βQ425 and βQ125), as reflected in the long-range crystallinity index (CrI_SR_) (Table 1). A notable change is observed in the XRD profiles of the colloidal substrates (βQ425CE and βQ125CE). Despite a considerable decrease in CrI_XRD_ values, there was a significant increase in the dimensions of the crystallites L_020_ and L_200_, indicating a relative increase in the crystalline domains dispersed throughout the matrix. Indeed, the profiles of βQ425CE and βQ125CE highlight diffraction peaks that are otherwise less evident in other β-chitin substrates. Furthermore, the average profile achieved by βQ425CE and βQ125CE resembles those of the substrates treated from α-chitins, as shown in (Figure 4c), especially in the emergence of peaks centered at 12.8^0^, 22.8^0^, and 26.5^0^, and the peak’s shift from 8.6^0^, to 9.4^0^, similar to what was observed for the spectral profile resulting from carbons C5 and C3 (Figure 3). The convergence of these trends suggests that the enzyme acts in favor of increasing the regularity and orderly packing of the molecular chains, although this trend cannot be deduced from the colloidal samples. In this case, αQCT25 and αQCT55 already consist of the structure of the more stable alpha polymorph, and there is not enough sensitivity in the diffractograms to indicate the influence of the enzymatic treatment.

### 3.4. Capillary Viscometry in Dilute Regime

To study the enzymatic activity on solid substrates and assess the enzyme’s behavior towards various substrates, reactions were conducted with solid substrates and the reaction medium was studied using the capillary viscometry technique. Table 2 shows that depolymerization of solid substrates occurred after enzymatic reaction (βQ425SE, βQ125SE, and αQSE), as the results of the intrinsic viscosity (η) and viscosity average molecular mass $\left(M_{v}\right)$ were lower than those for the solid substrates not treated with AsChtII-C4B1 (βQ425, βQ125, and αQ). Moreover, these results indicate that the enzyme’s action differs for each sample; there is about 4-fold and 2-fold reductions in $M_{v}$ for βQ125SE βQ425SE, respectively, after enzymatic treatment. This difference in $M_{v}$ possibly occurred because the βQ125 sample comprises smaller particles (i.e., a larger available surface area), facilitating the interaction between its surface and the enzyme. In contrast, the α-chitin sample treated with the enzyme (αQSE) exhibited a 12-fold reduction in its $M_{v}$, indicating a greater enzyme activity in this polymorphic form of chitin, possibly due to its more ordered structure.

The average degree of acetylation (GA) of the samples pre- and post-enzymatic treatment was calculated based on the results of the ^13^C NMR analysis. According to the results for solid substrates assessed without enzyme treatment, the results for solid substrates treated with the enzyme demonstrated that there was no significant change in GA.

### 3.5. Analysis of Enzyme Activity on the Growth of Fungus L. theobromae

Chitinases catalyze the degradation of α-chitin present in the shells of crabs and shrimps and in the cell walls of fungi [52], as well as catalyzing the degradation of α and β forms of chitin found in insects [53].

A previous study reported that the recombinant chitinase AsChtII-C4B1 demonstrated fungicidal activity against the filamentous fungus *Aspergillus fumigatus* [27]. In the present study, the enzyme was purified via ammonium sulfate precipitation and affinity chromatography and evaluated against the phytopathogenic fungus *L. theobromae*, an ascomycete belonging to class *Dothideomycetes*, order *Botryospheriales*, family *Botryosphaeriaceae* [54]. This phytopathogen attacks more than 500 species of plants, mainly in tropical and subtropical regions [55]. It causes fungal gummosis of peaches—a disease that severely restricts the the growth and production of this fruit in orchards in southern China, the United States, and Japan [56]. It causes leaf blight, stem cancer, and fruit rot in *Theobroma* cacao in Malaysia [57]. In Brazil, *L. theobromae* is a serious threat to cashew cultivation areas, causing resinosis and black rot of the stem [58]. In humans, this fungus has been associated with clinical manifestations such as corneal ulcers, rhinosinusitis, and mycosis in immunodeficient patients [59,60].

Given that *L. theobromae* grows optimally at 28 °C, the enzyme activity was tested at this temperature using colloidal α-chitin as a substrate, varying the reaction time. It was observed that the enzyme’s activity increased with reaction time, reaching a plateau at 48 h (Appendix A, Appendix A). Analysis of the enzyme’s thermostability showed that it retains 55% of its activity even after 72 h (data not shown), allowing for an evaluation of AsChtII-C4B1’s interference in the growth of this fungus in both solid and liquid media. Figure 5a–c show the growth of *L. theobromae* in a solid medium (Figure 5a) in the presence of a commercial fungicide (Figure 5b) and in the presence of the enzyme (Figure 5c). It was observed that the commercial fungicide *Amphotericin-B* completely inhibited mycelial growth, and the presence of the enzyme substantially inhibited this growth.

In liquid medium, the enzyme’s presence caused hyphae dispersion, suggesting that they were digested (Figure 5f). The fungus hyphae, both in the presence and the absence of the enzyme, were dried and weighed, and analyses showed a loss of mycelial mass after treatment with chitinase, as illustrated in Figure 6. Reductions in the mycelial mass of 20, 48.2, and 61.3% were observed after 24, 48, and 72 h, respectively, compared with the dry biomass of the fungus not subjected to the enzymatic process.

Although insects produce more chitinases than other organisms [61], most chitinases described in the literature with fungicidal activity are those from bacteria, fungi, and plants [62]. For example, the chitinase from *Trichoderma asperellum*, a fungus from the *Hypocreaceae* family, exhibited activity against the mycelium of fungus *Aspergillus nigers* [63], and the recombinant chitinase from cowpeas (*Vigna unguiculata*) inhibited the germination of spores and mycelial growth of the fungus *Penicillium herquei* [64].

The antifungal potential of chitinases depends on the morphology of the fungal cell walls, as the chitin contained in these microorganisms’ cell walls is associated with β-glucans 1/3 and 1/6 or other polysaccharides whose composition can vary according to the species [62,65]. Most fungal pathogens contain chitin, as a dominant component in the cell wall, varying in the range of 5 to 27% in dry mass. They are found mainly in the mitosis cycle of fungal cells and at the growing hyphal tip [5]. Thus, the fungicidal activity of a chitinase may present a specificity not only associated with the microstructure of the surface, but also with the proportion of chitin in the fungal cell wall.

To investigate the effect of AsChtII-C4B1 on the morphology of the chitin contained in the fungal cell wall, mycelium samples from experiments in a liquid medium were analyzed via microscopy. SEM images of fungus mycelium treated with recombinant chitinase show morphological changes in the hyphae (Figure 7). The fungus hyphae, in the absence of the enzyme, presented a dense network of long tubular structures with smooth surfaces (Figure 7a,b). The fungus hyphae treated with recombinant chitinase exhibited a disordered network and brittle, thinner tubular structures (Figure 7c). Other modifications observed in the hyphae included crushing of the tubular parts, long translucent hyphal tips, and holes on the surface (Figure 7d–l). It was also possible to observe the outer layers of the hyphae detaching from the fungal cell wall and to observe protruding, broken, and translucent tubular structures. Additionally, hyphae with wrinkled surfaces were observed in SEM images. Some cut hyphae, with rounded and brittle ends, were also observed. Therefore, these results demonstrate that the chitinase AsChtII-C4B1 inhibited the growth of phytopathogenic fungus *L. theobromae*, acting on the morphology of the hyphae both in the cell walls and in the degradation of nascent chitin corresponding to the hyphal tips, as demonstrated in SEM images.

Observations similar to those of this study were reported in pathogenic fungi affected by a recombinant bacterial chitinase (ChiKJ406136) [66]. Studies indicated that the mycelial cells of the fungus were ruptured, broken, and distorted. The antifungal activity of a recombinant plant chitinase (GlxChiB) also influenced the growth of the fungus *Trichoderma viride*, causing damage not only to the hyphal tips but also to the lateral cell walls [67].

Biological control using microorganisms that produce chitinases has favorable effects against many post-harvest fungal pathogens [68]. The application of chitinases in food preservation is promising, as they can degrade the cell wall of contagious fungi—one of the main problems in terms of food deterioration—and prevent the germination process of fungal spores, thus helping to reduce food decomposition and degradation [69].

## 4. Conclusions

Chitinases are enzymes of biotechnological interest because they play a key role in the degradation of polysaccharide chitin. The results presented here demonstrate that it is possible to produce GlcNAc from colloidal α- and β-chitin using a recombinant insect chitinase. AsChtII-C4B1 was capable of degrading various types of chitin substrates—an important characteristic for applications in biocatalytic processes such as the production of chitin derivatives. This study also investigated the fungicidal activity of the enzyme on the phytopathogenic fungus *L. theobromae*, which affects various economically important crops, causing a reduction in the quality and quantity of agricultural commodities. Chitinase reduced the biomass of the fungus and modified the structures of its hyphae, confirming the efficiency of chitinase in reducing fungal mycelial growth. The results shown in this work indicate that the chitinase AsChtII-C4B1 presents important characteristics for potential biotechnological applications and future studies need to be developed to optimize its properties so that it can effectively be used as a fungicide.

## Figures and Tables

**Figure 1 polymers-16-00529-f001:**
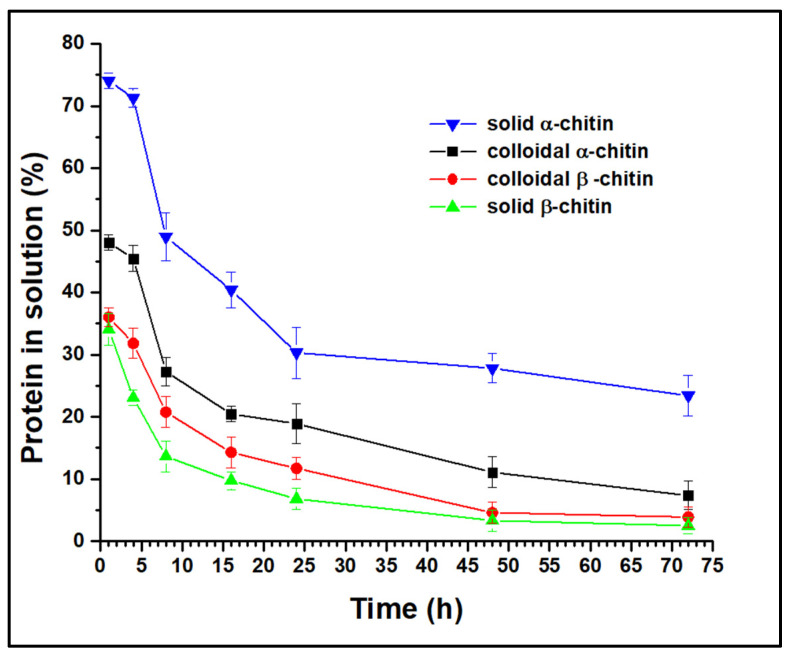
Analysis of free protein concentration in solution in reactions with α- and β-chitin substrates in both solid and colloidal forms. The experiments were performed in triplicate.

**Figure 2 polymers-16-00529-f002:**
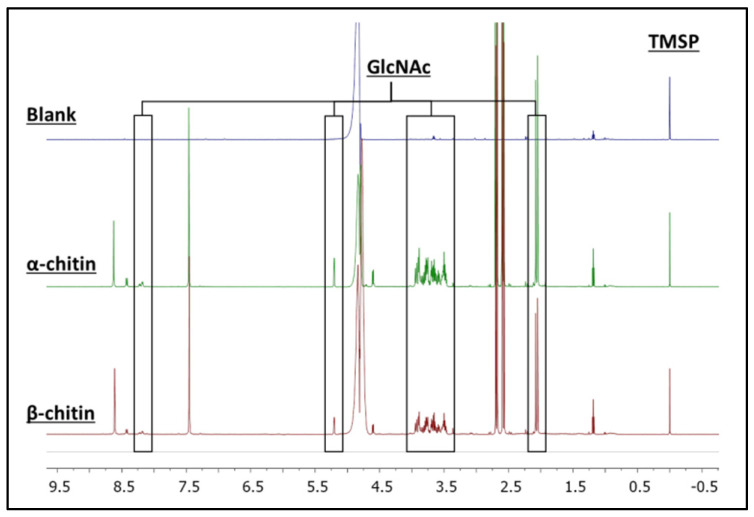
^1^H NMR spectra of the enzymatic reaction samples on colloidal α- and β-chitin. Signals of the hydrolyzed product, N-acetyl-d-glucosamine (GlcNAc), are highlighted. The concentration of the compound (GlcNAc) was determined from the signal of the internal standard (TMSP).

**Figure 3 polymers-16-00529-f003:**
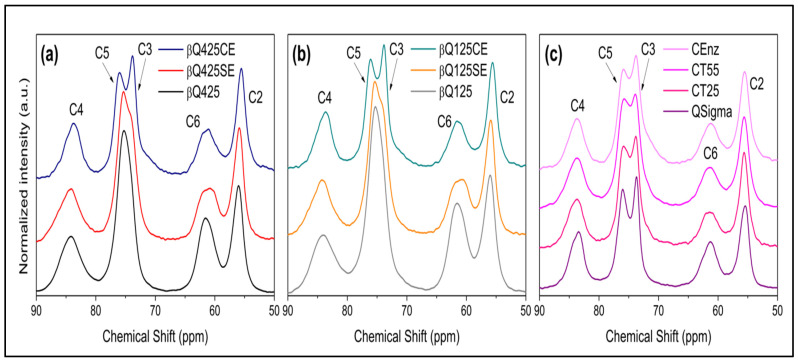
Structural characterization of chitin samples from the solid-state ^13^C NMR spectra: (**a**) βQ425: solid β-chitin, βQ425SE: solid β-chitin treated with the enzyme, and βQ425CE: colloidal β-chitin treated with the enzyme; (**b**) βQ125: solid β-chitin, βQ125SE: solid β-chitin treated with the enzyme, and βQ125CE: colloidal β-chitin treated with the enzyme; (**c**) Qsigma: solid α-chitin, QCT25, QCT55: colloidal α-chitin at varying temperatures of 25 and 55 °C, and QEnz: colloidal α-chitin treated with the enzyme.

**Figure 4 polymers-16-00529-f004:**
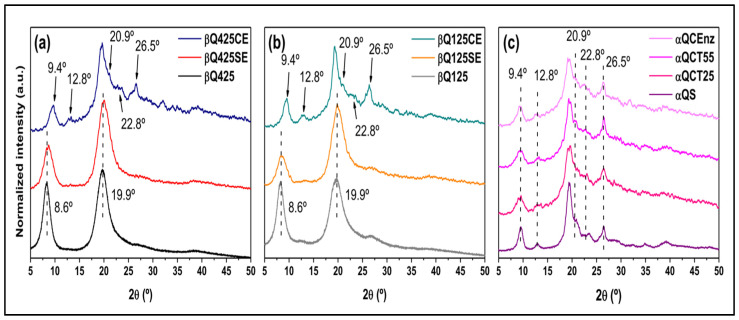
XRD patterns of β-chitin 125 (**a**), β-chitin 425 (**b**), and α-chitin (**c**) substrates.

**Figure 5 polymers-16-00529-f005:**
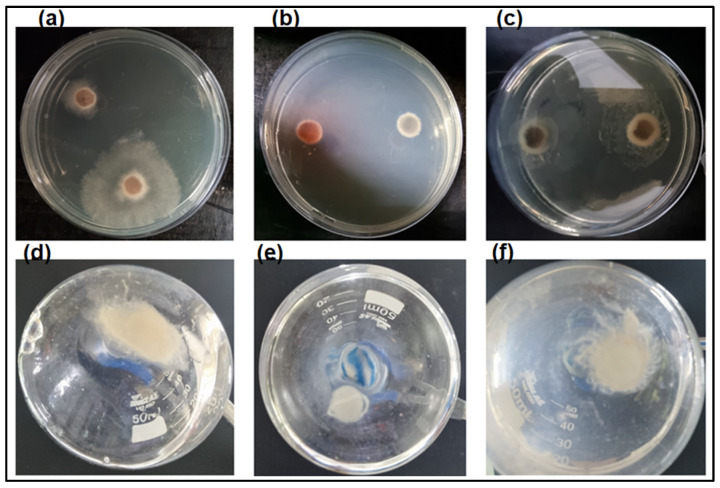
Chitinase activity on the growth of fungus *L. theobromae* in solid and liquid media: fungal growth (**a**) in the absence of the enzyme (positive control), (**b**) in the presence of commercial fungicide *Amphotericin-B* (negative control), and (**c**) in the presence of chitinase, conducted in Petri dishes containing solid medium (PDA); fungal growth (**d**) in the absence of the enzyme (positive control), (**e**) in the presence of commercial fungicide *Amphotericin-B* (negative control), and (**f**) in the presence of chitinase, conducted in liquid medium.

**Figure 6 polymers-16-00529-f006:**
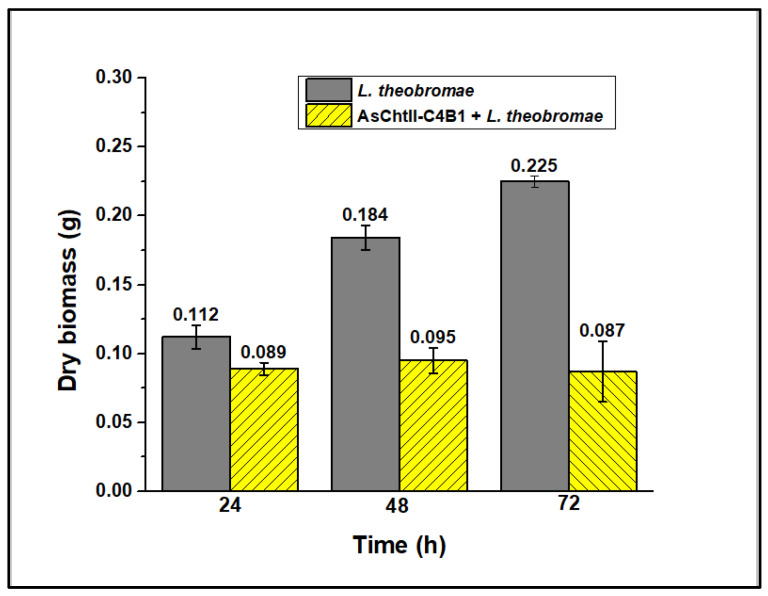
Evaluation of the dry mass of fungal mycelia before and after the action of chitinase. The assessment was conducted in three individual experiments varying the incubation time (24, 48, and 72 h). The experiments were performed in triplicate.

**Figure 7 polymers-16-00529-f007:**
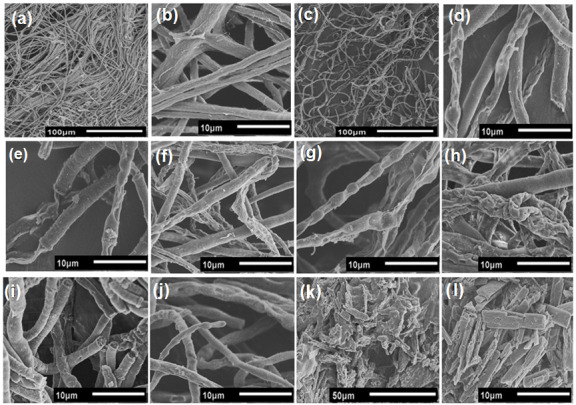
Morphological changes in the hyphae of fungus *L. theobromae* under the activity of the recombinant chitinase AsChtII-C4B1. (**a**,**b**): hyphae of the fungus in the absence of the enzyme. (**c**): hyphae of the fungus under the action of chitinase. (**d**–**l**): morphological alterations of the hyphae, such as crushing of the tubular parts, long translucent tips with holes on the surface, swollen tubes, rounded ends, and brittleness.

**Table 1 polymers-16-00529-t001:** Short-range (CrI_SR_) and long-range (CrI_XRD_) crystallinity indexes calculated via NMR and XRD, respectively; the dimensions of the crystallites correspond to the reflection planes (020)_h_ and (200)_h_.

Sample	CrI_SR_ (%)	CrI_XRD_ (%)	L_020_ (nm)	L_200_ (nm)
βQ425	83.8 ± 1.7	71.2	4.70	3.43
βQ425SE	75.7 ± 9.0	66.4	4.30	3.80
βQ425CE	79.0 ± 9.4	37.8	7.13	5.32
βQ125	80.7 ± 1.2	61.1	4.67	3.19
βQ125SE	76.3 ± 9.1	61.0	5.92	3.79
βQ125CE	79.5 ± 9.9	46.5	7.00	5.18
QSigma	85.7 ± 0.2	58.8	8.02	6.03
QCenz	75.1 ± 2.7	42.7	7.20	5.11

**Table 2 polymers-16-00529-t002:** Capillary viscometry results for the enzyme reaction medium with chitins: intrinsic viscosity (n)**,** viscosity average molecular mass ${(M}_{v})$ average degree of polymerization ${(GP}_{v})$, and average degree of acetylation (GA).

Sample	η (mL.mg^−1^)	$M_{v}$ × 10^6^ (g mol^−1^)	${GP}_{v}$	GA (%)
BQ 125	3.30 ± 0.22	996 ± 92	6187 ± 581	94.3 ± 4.35
BQ125SE	1.28 ± 0.02	253 ± 5.0	1465 ± 198	>95
BQ425	5.06 ± 0.74	1860 ± 39	11,546 ± 2400	90.3 ± 3.74
BQ425SE	3.03 ± 0.07	876 ± 30	4319 ± 148	>95
AQ	3.18 ± 0.18	943 ± 76.5	5858 ± 474	95.4 ± 5.28
AQ SE	0.57 ± 0.04	79.3 ± 7.8	492 ± 48	>95

## Data Availability

Data are contained within the article and Appendix A.

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
