# Peer review of "Activity of a Recombinant Chitinase of the Atta sexdens Ant on Different Forms of Chitin and Its Fungicidal Effect against Lasiodiplodia theobromae"

_polymers, 2024, doi:10.3390/polym16040529_

Round 1

Reviewer 1 Report

Comments and Suggestions for Authors

       The study presents the activity of a recombinant chitinase against colloidal and solid α- and β-chitin substrates. NMR, viscometry analysis were carried out to confirm the occurrence of hydrolysis reactions. The data in this paper are sound, I recommend it publication after the following issues is addressed:

1.     Some extra experiments should be carried out to explain why “α-chitin” is more preferable for AsChtII-C4B1 enzyme than “β-chitin”, what is mechanism?

2.     Besides viscometry, other more indicative characterization methods, such as GPC is suggested to monitor the hydrolysis reactions.

Minor issues:

1.     Page 1, line 7, “and*”

2.     Page 3, line 114, “2.2.1.1.α-”

3.     Page 6, line 255, “[28],”

4.     Page 10, line 375, ” 8.6º to 9.4º”

Comments on the Quality of English Language

none

Author Response

Reviewer #1: The study presents the activity of a recombinant chitinase against colloidal and solid α- and β-chitin substrates. NMR, viscometry analysis were carried out to confirm the occurrence of hydrolysis reactions. The data in this paper are sound, I recommend it publication after the following issues is addressed.

Thanks for your opinion about our work and about the manuscript.

  1. Some extra experiments should be carried out to explain why “α-chitin” is more preferable for AsChtII-C4B1 enzyme than “β-chitin”, what is mechanism?

The evaluator is right, other experiments must be developed to elucidate the mechanism of the enzyme's preference for alpha chitin. The results shown in this work served to evaluate the activity of the enzyme against different forms of chitin, and are studies that can be advanced in elucidating the mechanism of enzymatic action.

  1. Besides viscometry, other more indicative characterization methods, such as GPC is suggested to monitor the hydrolysis reactions.

Thank you for pointing this out. The SEC (Size Exclusion Chromatography) analyses would indeed be interesting, but we were not able to acquire the standards that allowed us to carry out the experiment and determine the average sizes of oligomers, enabling the assessment of the relative proportions of oligomers and monomers. However, such information would not have an impact on the investigation we conducted, which aimed to evaluate the activity of the recombinant enzyme on chitin substrates (both alpha and beta polymorphs). Therefore, the experiments we performed (intrinsic viscosity / average viscometric molecular mass) are sufficient and conclusive for determining the activity of the enzyme on the different substrates. 

Minor issues:

  1. Page 1, line 7, “and*”
  2. Page 3, line 114, “2.2.1.1.α-”
  3. Page 6, line 255, “[28],”
  4. Page 10, line 375, ” 8.6º to 9.4º”

We appreciate the observations and corrections, which have already been accepted and corrected in the text.

Reviewer 2 Report

Comments and Suggestions for Authors

Comments in attachment

Author Response

Reviewer #2: The manuscript is very valuable methodologically. However, I have concerns about the practical use of the research findings. This research should be considered as basic research. Chitinases can only be applied after immobilisation on a suitable support. These are enzymes that lose their activity very quickly and become useless. The authors should address this issue in their conclusions and indicate that this is preliminary research. Further research should focus on developing a way to preserve the activity of chitinases.

We are grateful for the topics pointed out by the referee and the interest shown.

We added a sentence in the topic ´Conclusions´ reinforcing that our studies are preliminary and that more studies are necessary for an efficient application of the enzyme as a fungicide, for example.

´The results shown in this work indicate that the chitinase AsChtII-C4B1 presents important characteristics for potential biotechnological application and future studies need to be developed to optimize its properties so that it can effectively be used as a fungicide. ´

Some comments:

- Are the authors sure that chitinases inhibit fungal growth ? The enzymes were not purified, they did not test the specificity of chitinases. According to many researchers, only endochitinases inhibit fungal growth.

The enzyme tested in the fungus growth experiments was purified by precipitation with ammonium sulfate followed by affinity chromatography on nickel resin, as described in MICOCCI et., 2023. Therefore, the enzyme was above 90% pure, as verified by electrophoresis. We did not cite this data in the previous text, but we added this information in the new text (page 11, line 452)

- The authors should make it clear in the text that their research has application potential after accurate formulation.

We added a sentence in the topic ´Conclusions´ reinforcing that our studies are preliminary and that more studies are necessary for an efficient application of the enzyme as a fungicide, for example.

´The results shown in this work indicate that the chitinase AsChtII-C4B1 presents important characteristics for potential biotechnological application and future studies need to be developed to optimize its properties so that it can effectively be used as a fungicide. ´

- In the Materials and Methods chapter, in line 218, instead of "biological activity" write "antifungal activity". The term 'Biological activity' is very wide and can mean many parameters, e.g. the production of antibiotics or growth promoters.

The reviewer raised an interest point, and we are grateful for the observation.